# Critical Review of Tuberculosis Diagnosis in Children from Papua New Guinea Presenting to Health Facilities in the Torres Strait Islands, Australia

**DOI:** 10.3390/microorganisms11122947

**Published:** 2023-12-08

**Authors:** J’Belle Foster, Ben J. Marais, Diana Mendez, Emma S. McBryde

**Affiliations:** 1College of Medicine and Dentistry, James Cook University, Townsville, QLD 4811, Australia; jbelle.foster@my.jcu.edu.au; 2Australian Institute of Tropical Health and Medicine, James Cook University, Townsville, QLD 4811, Australia; diana.mendez@jcu.edu.au; 3Torres and Cape Tuberculosis Control Unit, Thursday Island, QLD 4875, Australia; 4WHO Collaborating Centre in Tuberculosis, Sydney Infectious Diseases Institute (Sydney ID), The University of Sydney, Westmead, NSW 2145, Australia; ben.marais@health.nsw.gov.au

**Keywords:** paediatric tuberculosis, Torres Strait Islands, tuberculosis diagnostics, Torres Strait/PNG border, WHO Paediatric TB Algorithm, The Union Desk Guide, cross border

## Abstract

Paediatric tuberculosis can be challenging to diagnose, and various approaches are used in different settings. A retrospective review was conducted on Papua New Guinea (PNG) children with presumptive TB who presented for health care in the Torres Strait Islands, Australia, between 2016 and 2019. We compared diagnostic algorithms including the modified Keith Edwards TB Score, The Union Desk Guide, and the new World Health Organization (WHO) algorithm, with diagnostic practices used in the remote Torres Strait Islands. Of the 66 children with presumptive TB, 7 had bacteriologically confirmed TB. The majority (52%) were under 5 years (median age 61 months), and 45% were malnourished. There was moderate agreement across the diagnostic methods (K = 0.34; 95% CI 0.23–0.46), with the highest concordance observed between The Union Desk Guide and the WHO’s algorithm (K = 0.61). Local TB physicians might have over-diagnosed presumed lymph node TB while under-diagnosing TB overall. Enhancing the precision and promptness of paediatric TB diagnosis using practical tools is pivotal to decrease TB-related child mortality, notably in isolated regions like the Torres Strait and the Western Province of PNG.

## 1. Introduction

Tuberculosis (TB) is recognised as a top ten cause of under-5 mortality in TB endemic areas, such as Papua New Guinea (PNG) [1]. Globally, approximately 10% of all TB cases diagnosed are in children aged <15 years [2], but in Papua New Guinea, this is around 30%. While children with TB are generally less contagious than adults, their mortality risk is higher, especially in the very young and immunocompromised [3]. Active TB disease typically develops in children within the first 12 months after exposure and primary infection [4]. An estimated 90% of children who die from TB, die without ever being diagnosed or accessing TB treatment and care [5].

Persistent gaps in TB case detection in children result from the pauci-bacillary nature of their disease and their inability to expectorate sputum [6]. Microbiological confirmation takes up to six weeks, yields are low and specimen collection is difficult. In the absence of more advanced diagnostic options, pragmatic diagnostic approaches such as scoring tools and algorithms may assist with diagnosis in children and improve their access to treatment and care [7].

PNG residents from Treaty villages in the Western Province of PNG can access basic diagnostic and referral services via Australian health facilities in the Torres Strait (Map: doi. 10.6084/m9.figshare.16632823, accessed 4 October 2023). Typically, bacteriologic studies available for these children are limited to voluntary expectoration, nasopharyngeal aspirates in outpatient settings, nasogastric aspirates in inpatient settings, and rarely, sputum induction. Once children report to a Primary Health Centre in the Torres Strait, clinical evaluation undertaken by Advanced Health Workers and nurses is triggered if minimum criteria are met (Figure 1). Once diagnosed with TB in the Torres Strait, PNG patients must then access TB services located on Daru Island in the Western Province of PNG, to commence treatment. In PNG’s Western Province, the estimated TB incidence rate is very high (736/100,000 in 2017) [8], and nearly a third (27%) of TB case notifications are in children < 15 years of age [9].

To date, no validated TB diagnostic tool or algorithm has been used in the Torres Strait to assist TB triage or diagnosis in children. The use of symptom-based clinical diagnostic approaches in remote settings are supported by both the World Health Organization (WHO) and the International Union against Tuberculosis and Lung Disease (The Union) [10,11] to reduce TB related mortality and enable children to enter the TB care pathway sooner. We aimed to compare current paediatric TB diagnostic approaches used in the Torres Strait with three internationally recognised approaches: (1) the Keith Edwards TB Score developed in PNG (1987) (Appendix A) [12]; (2) The Union Desk Guide, 3rd edition (2016) (Appendix A) [10]; and (3) the new WHO evidence-based paediatric TB algorithm (2022) (Appendix A) [13].

## 2. Materials and Methods

We performed a retrospective study including all paediatric patients (n = 66) (aged 0–14 years) from PNG that presented with presumptive TB to a Queensland Health facility in the Torres Strait Protected Zone between January 2016 and December 2019. TB case definitions included bacteriologically confirmed or clinically diagnosed cases [14]. Presumptive TB refers to a clinical scenario where a patient is suspected of having TB based on presenting signs, symptoms, and epidemiological factors, but where a formal diagnosis —either laboratory confirmation or clinical—has not yet been obtained. Presumptive TB necessitates a careful clinical approach before a decision on initiating TB treatment is made. The treatment decision is often based on a combination of clinical judgment and the use of diagnostic tools like TB scoring systems, which take into account various signs and risk factors to estimate the likelihood of TB.

Sites of TB disease included pulmonary, extra-pulmonary, or both. The study excluded children who resided in the Torres Strait, since their data was captured by a separate system for Australian residents.

Diagnostic work-up of PNG nationals at health facilities in the remote Australian Torres Strait Islands is usually triggered when a child presents with at least one sign or symptom suggestive of TB [14]. Minimum criteria for TB work-up includes a cough for >2 weeks and/or night sweats, fever of unknown origin, unexplained weight loss, enlarged lymph nodes (>1 cm, >1 month, +/− discharging sinus), or haemoptysis. Observations and notification of these signs and symptoms were recorded on an ‘Initial Visit’ form provided by the local TB Unit. TB work-up and specimen (predominantly sputum) collection is performed by Primary Health Centre clinicians who, in close consultation with a rural General Practitioner (GP), provide a dedicated outreach service [14].

Patient charts and data on all presumptive TB cases among PNG nationals contained in the electronic database used in the Torres and Cape Hospital and Health Service, called Best Practice, were accessed, as were ‘Initial Visit’ forms. Chest X-ray (CXR) results were obtained from Queensland Health’s radiological software (PACS) and pathology results from Queensland Health’s laboratory information system called AUSLAB. 

Weight loss was based on subjective reports of recent loss of weight by patient’s parents or care givers. The WHO online weight-for-age calculator was used to manually calculate weight-for-age percentiles for the purpose of this study. The notation ≤ 0.00% refers to cases where the children’s weight-for-age Z-score was more than 2 standard deviations below the mean, thus indicating severe malnutrition.

### Diagnostic Approaches

1. Clinical decision made by local physician: Diagnostic practices in the region were based on the presence of clinical features of disease and epidemiological risk factors, and the results of diagnostic investigations following the clinical algorithm in Figure 1 [15].

At the Australia/PNG border, local policies can influence the diagnostic process, including the selection of diagnostic methods and the management and follow-up of patients. Unfortunately, local policy prevents recalling PNG patients across the international border for any purpose, including tuberculin skin test (TST) reading, repeat CXR, or review following a course of antibiotics or nutritional rehabilitation [16]. Health services rendered to presumptive TB patients who are in a ‘non-critical’ condition, are provided with the specific purpose of identifying TB and referring patients back to the PNG health system for ongoing care [17]. Australia has made significant investments to support quality TB care on Daru Island, which is on the PNG side of the border [18].

2. Keith Edwards TB Score: The Keith Edwards TB Score (Appendix A) was developed in PNG and a score of ≥7 indicates a high likelihood of paediatric TB [12,19]. The Keith Edwards TB Score exemplifies a practical approach to diagnosing TB in children, especially in settings where laboratory resources are scarce. The allocation of points in the score is based on the presence and severity of symptoms and TB risk factors. For example, more points might be given for a longer duration of cough, while TB risk factors include a history of close contact with a sputum smear-positive TB case. Two data points included in the Keith Edwards TB Score, namely TST reading and malnutrition with failure to improve after four weeks of nutritional rehabilitation (collectively worth six points), were not collected in our setting and had to be excluded from this study.

Modifications were made to adapt the Keith Edwards TB Score (1987) to the available data and the practical realities of diagnosing TB in children in the Torres Strait region (Appendix A). The original Keith Edwards TB Score allocates points based on nutritional status, with different points for patients >80%, 60–80%, and <60% of the expected weight for age [19]. In this study, the WHO weight-for-age percentile charts and the WHO weight-for-age calculator were used to adjust these criteria. Modification involved adapting the scoring to align with WHO standards for assessing weight-for-age [20]. We used WHO weight-for-age percentile charts up to 10 years of age, and the WHO weight-for-age calculator, and adjusted the criteria as follows: no points if ≥15th percentile, one point if 3–14th percentile, and three points if <3rd percentile. Further, no points are allocated in the Keith Edwards TB Score to patients with no family history of TB, one point for contact with sputum smear-negative TB and three points for contact with sputum smear-positive TB [19]. As patients presenting to health facilities in the Torres Strait rarely knew if TB source cases were sputum smear-positive or -negative, we allocated two points for known TB contact, irrespective of sputum smear status. When the duration of fever or night sweats was recorded as ≤2 weeks, no points were allocated, and 2 points were allocated if >2 weeks.

3. The Union Desk Guide: The Union Desk Guide (2016) (Appendix A) requires persistent signs and symptoms of TB to be present [10]. Patients with signs and symptoms of ≤2 weeks were excluded from a possible positive diagnosis and enlarged lymph nodes were allocated one point (Appendix A). The Union Desk Guide recognises microbiologically confirmed and clinically diagnosed cases. A clinical TB diagnosis requires at least two of three features to be present, where one point is allocated against each of the following: (1) close contact with a known case, (2) signs and symptoms of TB, and (3) a CXR suggestive of TB. Therefore, the maximum number of points that could be allocated in this study was four, inclusive of laboratory confirmation. 

All CXR reports written by a radiologist were reviewed and a point allocated for any parenchymal opacification or cavity, miliary lesions, intrathoracic lymphadenopathy or when the radiologist indicated that the CXR was suspicious for TB. Where the radiologist indicated an abnormality in the report but where there was limited description, a TB Specialist (B.J.M. or C.C.) reviewed and reported on whether the abnormality was TB-related.

4. New WHO Paediatric TB Algorithm: The WHO recently published consolidated new evidence-based guidelines on the treatment and care of TB in children [21], in conjunction with a detailed operational handbook [13]. The operational handbook contains a new evidence-based algorithm to improve TB case detection in high burden settings [13]. The algorithm (Appendix A) focuses on practical treatment guidance in children <10 years with presumptive pulmonary TB and accepts relatively low specificity to ensure a sensitivity of at least 85%. Entry into the algorithm requires the presence of suspicious pulmonary signs and symptoms for a minimum duration of 1–2 weeks. Vulnerable children, defined as those aged <2 years, with severe acute malnutrition and/or human immunodeficiency virus (HIV), do not have to meet minimum criteria for entry into the algorithm if they are contacts of known cases (Appendix A). Past this point, the algorithm then allocates points to other suspicious signs and symptoms including enlarged lymph nodes and suggestive CXR changes (if CXR is available) with a score >10 indicating a need for treatment.

Modifications made to the new WHO algorithm were necessitated by data and study setting limitations and are reflected in Appendix A. The new WHO algorithm typically requires that some ‘low risk’ patients be followed up within 1–2 weeks to assess for persistent or worsening symptoms. This aspect had to be adapted to accommodate the ‘no recall’ policy in the study context.

Apart from high-risk patients (aged <2 years and <3rd percentile for age/weight), children required >2 weeks of signs and symptoms to be assessed. Due to the lack of recorded data on the time since exposure to a known TB case, the study assessed any close contact of a known TB case. This differs from the standard WHO algorithm, which focuses on contacts exposed in the past 12 months.

In the absence of height and growth trajectory data, severe acute malnutrition in the study was defined based on a weight-for-age Z-score more than 2 standard deviations below the mean. This corresponds to a weight-for-age percentile of ≤0.00%. Patients with weight <3rd percentile for age were considered to have a suggestive sign of TB [13].

Analyses: Frequencies and proportions were calculated using SPSS (version 28; New York, NY, USA, 2021). The Pearson’s chi square test was used to determine the association between diagnostic outcomes and select individual and clinical characteristics. 

Fleiss’ kappa was used to assess agreement between the four different TB diagnostic approaches evaluated. Standard kappa tests were used to measure individual agreement between each of the approaches. 

Ethical clearance and a waiver of consent were obtained from Far North Queensland Human Research Ethics Committee (HREC/17/QCH/74-1157), and James Cook University (H7380). All data were obtained with approval from data custodians within Queensland Health and with Public Health Act authorisation (QCH/36155–1157).

## 3. Results

Table 1 reflects the demographics and other characteristics of the 66 PNG children, aged between 3 months and 14 years, who presented with presumptive TB to an Australian health facility in the Torres Strait. Of these, 21 (32%) were diagnosed with TB by local TB physicians. Most children (34/66; 52%) were <5 years of age, including 12/21 (57%) diagnosed with TB. Of 65 children in whom a weight was recorded, 29/65 (45%) were <3rd percentile for age. Using WHO weight-for-age percentiles in 10 patients diagnosed with TB, 90% (9/10) fell beneath the third percentile and were identified as having severe acute malnutrition (as defined).

No child tested HIV positive (0/27 tested). Only one child had received a Bacille Calmette Guérin (BCG) vaccine. The most common presenting symptoms were fever (54/66; 82%) and cough (47/66; 71%). Of those with a cough, 25/47 (53%) coughed for >2 weeks. Of the 21 patients treated for TB, 7/21 (33%) had microbiological confirmation; two with confirmed multidrug-resistant (MDR) TB. None of the six patients diagnosed with lymph TB had a fine needle aspiration performed.

Table 2 shows the level of agreement between the four diagnostic approaches evaluated. Overall, there was fair agreement between the specified diagnostic approaches (K = 0.34; 95% Confidence Interval [CI] 0.23–0.46), with the best agreement between The Union Desk Guide and the new WHO algorithm (K = 0.61; *p* < 0.001). Figure 2 reflects individual patient level agreement between the different diagnostic approaches, demonstrating variable agreement.

Figure 2 and Figure 3 provide a detailed description of individual features documented in children with presumptive TB. The modified Union Desk Guide correctly identified all seven cases with microbiologically confirmed TB, compared to the modified Keith Edwards TB Score that correctly identified four cases. Of the five cases aged <10 years that were diagnosed with microbiologically confirmed TB, four were identified as requiring TB treatment by the new WHO algorithm. The remaining case did not have persistent symptoms suggestive of pulmonary TB and was not considered a ‘vulnerable child’ and therefore did not progress through the diagnostic pathway.

More paediatric TB cases were diagnosed using the Keith Edwards TB Score, (26), the Union Desk Guide (29) and the new WHO algorithm (27), than were diagnosed by local TB physicians (21). When applying the modified scores, 19 (Keith Edwards TB Score), 15 (Union Desk Guide), and 14 (new WHO algorithm) cases would have been diagnosed. Of six lymph node TB cases diagnosed by a local TB physician, consensus was only achieved across all diagnostic approaches for one case, acknowledging that the new WHO algorithm specifically focuses on pulmonary TB in children. 

The TB threshold in Figure 3 acts as a clinical guide to initiate treatment based on a calculated likelihood of the disease, balancing the risks of under-treatment and over-treatment. It should be noted that while the TB threshold is a useful tool, it is not a substitute for laboratory confirmation where possible. It is primarily a means to ensure timely treatment in the face of diagnostic uncertainties and constraint. In Figure 3, the term “TB threshold” is used for both diagnostic purposes and treatment initiation in a context in which patients do not typically commence effective TB treatment in Australia.

## 4. Discussion

Overall, our study demonstrated that the new WHO algorithm was more inclusive than other diagnostic approaches, although one culture-confirmed case was not detected. In the absence of a defined reference standard, we are not able to comment on diagnostic accuracy, but the highest agreement was observed between The Union Desk Guide (2016) and the new WHO algorithm. Based on the findings of the study, use of the new WHO algorithm complemented by the Union Desk Guide seems like the most inclusive approach, but more data are required, from other settings, to assess feasibility and accuracy. Diagnostic criteria need to be sensitive to local epidemiological factors and resource constraints, with emphasis on the need to assess nutritional status. In such remote settings, the promptness of diagnosis is critical for timely commencement of effective treatment. 

There is an urgent need to improve children’s access to TB treatment, especially in high TB incidence settings, to reduce TB-related mortality [22,23,24]. Although some overtreatment is preferable in some high TB incidence settings to under-treatment, clinicians in the Torres Strait need to find a difficult balance between timely TB diagnosis and not referring cases to the PNG health system unnecessarily [14]. Although case notifications incur cost [25] and children rarely pose a transmission risk to the community, the treatment of all children with TB is essential to save lives.

We had to modify each tool to fit with available diagnostic infrastructure in the Torres Strait, but our modifications generally led to lower scores; therefore, fewer patients reaching threshold for diagnosis than might otherwise have been expected. Despite these conservative modifications, more cases were diagnosed using each of the three diagnostic approaches than those diagnosed by local TB physicians. Each of the algorithms could be applied in the Torres Strait Islands, but restricted follow up and limited laboratory diagnostic support would hamper optimal application. CXR is readily available on two Australian islands close to the PNG border, but the quality and cost are variable, while wide interobserver variability limits its use as a reference standard [6]. Nasopharyngeal or nasogastric aspiration and induced sputum collection is not always available [26] and is dependent upon the skillsets of local nurses and clinicians. Similarly, specimens required for an extra-pulmonary TB diagnosis such as pleural fluid or lymph node aspirates are not typically collected in the Torres Strait. In this study, more than half of all cases diagnosed locally had extra-pulmonary TB involvement, and of those, 55% were diagnosed with cervical TB lymphadenitis.

Enlarged cervical lymph nodes due to TB can be confused with many other causes, including cancer [27], which is why fine needle aspiration biopsy (FNAB) is such a valuable tool to confirm the correct diagnosis and ensure appropriate treatment. FNAB of cervical lymph nodes that are persistent and greater than 1 × 1 cm is an underutilized, but safe and simple diagnostic modality that can enhance diagnostic accuracy [28]. The sensitivity and specificity of FNAB to establish a TB diagnosis is excellent [29] and yields with Xpert MTB/RIF^®^ (Sunnyvale, CA, USA) or Ultra^®^ are good [28]. Training and implementation of FNAB and access to Xpert MTB/RIF^®^ or Ultra^®^ could improve diagnostic accuracy [30], but the time delay in establishing a diagnosis if specimens need to be sent away remains problematic when dealing with cross-border patients. 

According to PNG health statistics, 24% of children are underweight and 14% experience moderate or severe forms of malnutrition [31]. These figures may reflect poor food security combined with poor sanitation and hygiene [32].

Both the new WHO algorithm and the Union Desk Guide work optimally with access to longitudinal growth data to ascertain ‘failure to thrive.’ ‘Failure to thrive’ is a measure of inadequate weight gain and growth over time [33], and as there is a no recall policy in place at the Torres Strait/PNG border, identifying ‘failure to thrive’ or monitoring weight over time is not possible.

WHO age-for-weight percentiles and anthropometric measurements such as weight-for-height (BMI) and middle upper arm circumference (MUAC) are not routinely calculated for paediatric patients presenting with presumptive TB in the Torres Strait. Low BMI and MUAC have been shown to be predictors of both TB mortality [34] and TB disease, and both have shown to improve with TB treatment [35]. 

A child with a BMI Z-score of >3 standard deviations below the mean of WHO growth standards, or a MUAC < 115 mm (6–59 months) or <130 mm (5–9 years) indicates severe acute malnutrition with increased mortality risk [36,37]. Of the 29 children who presented with malnutrition, TB was diagnosed in 10 patients, demonstrating the high yield of TB detection among severely malnourished children in TB endemic settings. Similar findings have been reported in African studies [38,39,40]. Where high rates of TB and severe acute malnutrition coexist, TB screening should always include a malnutrition assessment. A shift in policy to enable these diagnostic pathways, in the same way TB screening automatically follows an HIV diagnosis and vice versa [24] is indicated and is likely to improve the management and outcomes of paediatric TB in these regions.

During the period of our study, it is important to note that the BCG vaccine was not routinely available to PNG nationals. The inconsistent delivery of BCG vaccination underscores the importance of developing and implementing effective diagnostic and treatment strategies for TB, particularly in vulnerable young children. Together with improved uptake of BCG vaccination, improved strategies to address malnutrition may be a valuable primary TB prevention strategy in this setting. In the interim, routine monitoring of nutritional status and key malnutrition datapoints such as weight and MUAC should be incorporated into existing policies and procedures to provide critical triage points [41] for improved TB screening [42].

Some important study limitations require consideration. As described above, due to the retrospective nature of the study, all three of the diagnostic algorithms used in this study needed to be modified. This is because TB management in the Torres Strait is limited by borderlands regulation, high staff turnover limiting the ability to implement a consistent and sustainable source of TST providers in remote primary healthcare settings, and a lack of advanced point of care diagnostic technology such as the Xpert MTB/RIF^®^ or Ultra^®^. The small sample size limited our ability to provide a comprehensive overview and assess the significance of findings, but it facilitated a more detailed description of the study cohort. A strength of the study is that it included all patients that presented with signs and symptoms of TB during the study period. Unfortunately, we were unable to report on local mortality risks associated with TB and malnutrition, which highlights the difficulty of cross-border care delivery.

## 5. Conclusions

Diagnosing paediatric TB in remote settings without the possibility of recall across an international border is challenging. High rates of malnutrition and low BCG vaccination coverage are major concerns in paediatric patients from PNG presenting to health facilities in the Torres Strait.

Reducing the likelihood of missed diagnoses and time to effective treatment commencement requires further exploration of optimal diagnostic approaches in this remote setting. Implementing modified algorithms that are suitable in this context, as part of an ongoing prospective operational research project, will be useful, as would the experience from other remote settings that experience similar challenges.

## Figures and Tables

**Figure 1 microorganisms-11-02947-f001:**
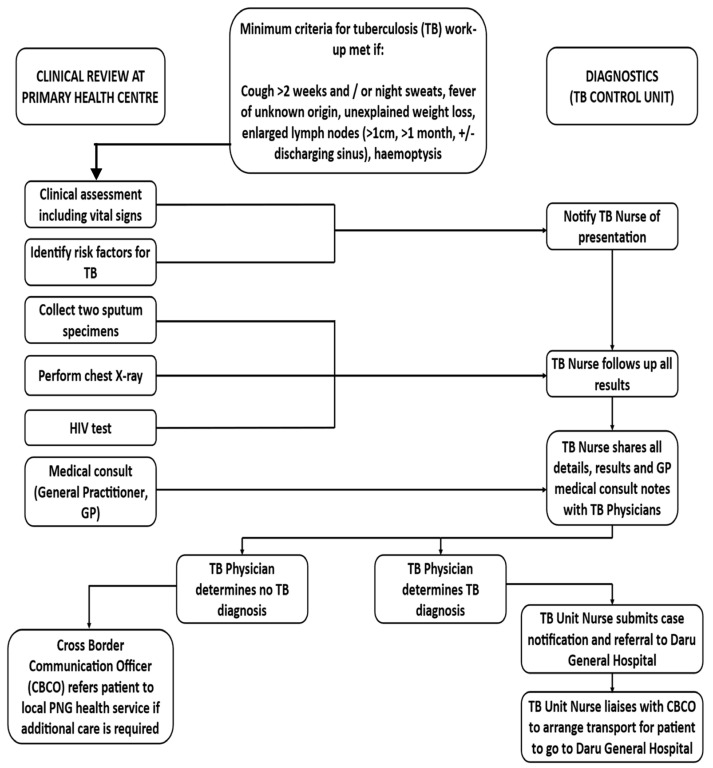
Diagnostic work-up for patients from Papua New Guinea visiting health services in the Torres Strait Protected Zone.

**Figure 2 microorganisms-11-02947-f002:**
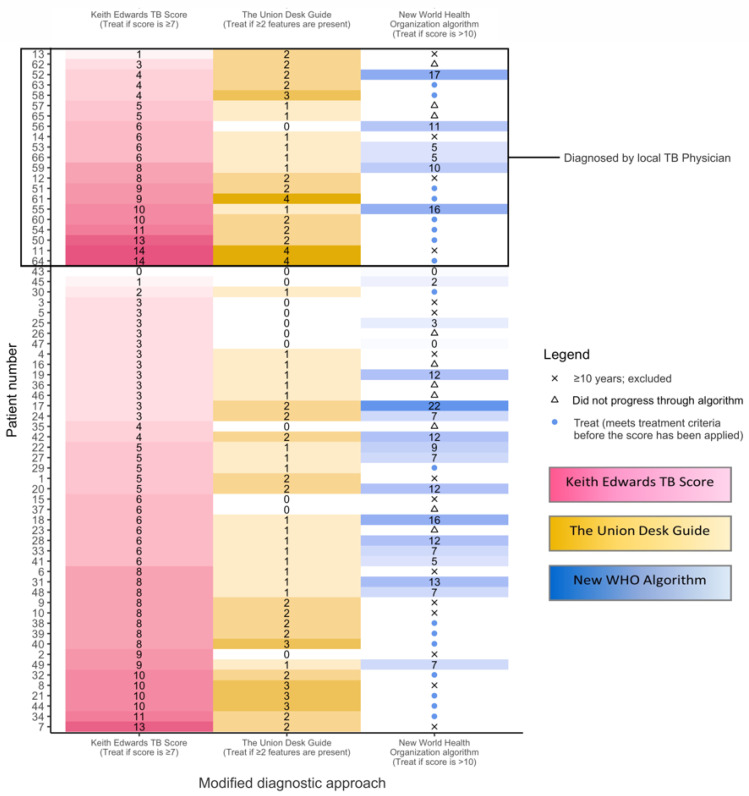
Illustration of individual level agreement between local TB physicians’ diagnoses, and the modified Keith Edwards TB score, The Union Desk Guide, and the new World Health Organization algorithm. Note: The colours used relate to different tools used. This is a ‘heatmap’ of the scores obtained for each patient with each tool—the higher the score, the darker the shade.

**Figure 3 microorganisms-11-02947-f003:**
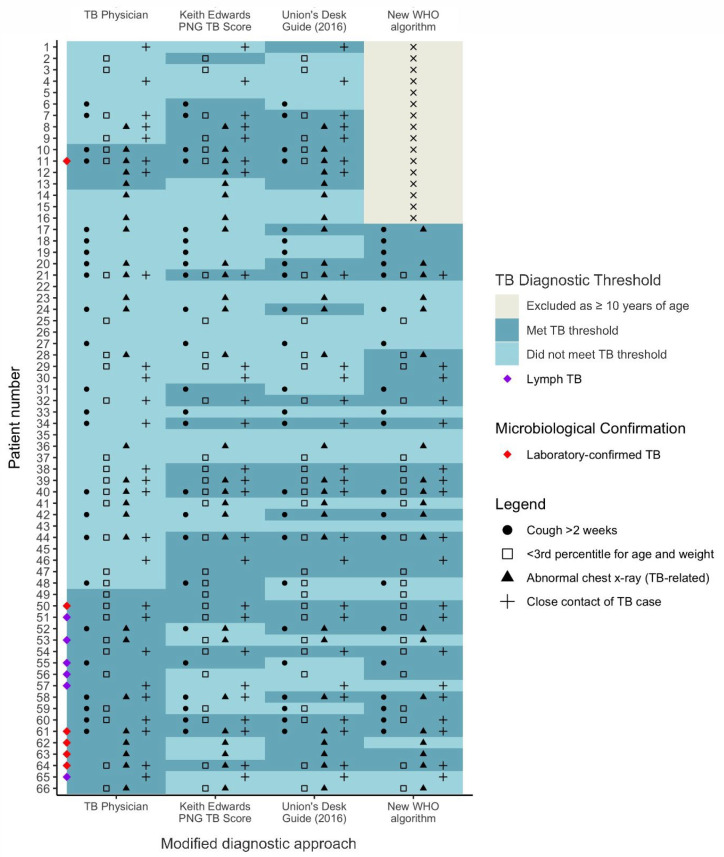
Detailed description of individual features documented in children with presumptive TB comparing local Physicians’ diagnoses, and the modified Keith Edwards TB score, The Union Desk Guide, and the new World Health Organization algorithm. PNG, Papua New Guinea; TB, tuberculosis; WHO, World Health Organization. Note. TB threshold refers to ‘guidance to treat for TB’; not microbiological confirmation or any other objective reference standard.

**Table 1 microorganisms-11-02947-t001:** Characteristics of Papua New Guinea children who presented to health facilities in the Torres Strait Protected Zone with presumptive tuberculosis (2016–2019).

Characteristic	TB Treatment Advised N = 66 (%)		
	Yes	No	Total
**Age group (median 61 months)**			
<5 years	12 (35)	22 (65)	34 (52)
5–9 years	5 (29)	12 (71)	17 (26)
10–14 years	4 (27)	11 (73)	15 (23)
**Sex**			
Female	9 (29)	22 (71)	31 (47)
Male	12 (34)	23 (66)	35 (53)
**Visa Status**			
Papua New Guinea Treaty Visitor	17 (30)	39 (70)	56 (85)
Papua New Guinea non-Treaty Visitor	4 (40)	6 (60)	10 (15)
**Primary Health Centre Attended**			
Saibai	18 (33)	37 (67)	55 (83)
Boigu	3 (33)	6 (67)	9 (14)
Other	0 (0)	2 (100)	2 (3)
**Recent close contact of a known TB case**			
Close contact	11 (42)	15 (58)	26 (39)
No close contact	10 (25)	30 (75)	40 (61)
**Chest X-ray (CXR) ^˄^**			
CXR performed	18 (34)	35 (66)	53 (80)
CXR not performed	3 (23)	10 (77)	13 (20)
**Nutritional status**			
<3rd percentile for age and weight	10 (34)	19 (66)	29 (44)
Severe acute malnutrition	9 (90)	7 (37)	16 (55)
**TB signs and symptoms ***			
Cough >2 weeks	7 (28)	18 (72)	25
Fever	14 (26)	40 (74)	54
Night sweats	7 (37)	12 (63)	19
Weight loss	10 (33)	20 (67)	30
Enlarged lymph nodes^#^	11 (44)	14 (56)	25
Haemoptysis	1 (100)	0 (0)	1

^˄^ CXR—chest radiograph. Note. * Signs and symptoms as reported by the parent or care giver, each patient may have more than one sign or symptom recorded; ^#^ defined as >1 cm × 1 cm for >1 month, +/− discharging sinus.

**Table 2 microorganisms-11-02947-t002:** Individual agreement between TB physician diagnosis, the Keith Edwards TB score, The Union Desk Guide, and the new World Health Organization algorithm^#^ for the diagnosis of paediatric TB.

		New World Health Organization Algorithm				Union Desk Guide				Keith Edwards TB Score			
		Not TB	TB *	Total	K	Not TB	TB *	Total	K	Not TB	TB *	Total	K
**Local TB Unit**	**Not TB**	18	16	34	0.15	29	16	45	0.24	29	16	45	0.11
	**TB**	6	11	17		8	13	21		11	10	21	
	**Total**	24	27	51		37	29	66		40	26	66	
**New WHO algorithm**	**Not TB**	NA	NA	NA	NA	22	2	24	0.61	21	3	24	0.42
	**TB**	NA	NA	NA	NA	8	19	27		12	15	27	
	**Total**	NA	NA	NA	NA	30	21	51		33	18	51	
**Union Desk Guide**	**Not TB**	22	8	30	0.61	NA	NA	NA	NA	30	7	37	0.47
	**TB**	2	19	21		NA	NA	NA	NA	10	19	29	
	**Total**	24	27	51		NA	NA	NA	NA	40	26	66	

Overall agreement (K = 0.34; 95% Confidence Interval [CI] 0.231–0.46; *p* < 0.001). K: Cohen’s kappa value; TB: tuberculosis; WHO: World Health Organization. * TB refers to ‘guidance to treat for TB’, not microbiological confirmation or any other objective reference standard. ^#^ The New WHO algorithm excludes children aged ≥10 years, and hence, these children were excluded from Table 2.

## Data Availability

As tuberculosis is a notifiable disease in Queensland, Australia, public sharing of data is restricted due to confidentiality clauses. Access to data requires Human Research Ethics Committee, Public Health Act, and Site-Specific Access approvals via Queensland Health.

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
