# Peer review of "Critical Review of Tuberculosis Diagnosis in Children from Papua New Guinea Presenting to Health Facilities in the Torres Strait Islands, Australia"

_microorganisms, 2023, doi:10.3390/microorganisms11122947_

Round 1

Reviewer 1 Report

Comments and Suggestions for Authors

Thank you for the opportunity to help review this manuscript entitled “Critical review of tuberculosis diagnosis in children from Papua New Guinea presenting to health facilities in the Torres Strait Islands, Australia. This retrospective study by Foster et al, investigated and compared current diagnostic algorithms with local protocol for diagnosing TB in children. This is an interesting study that provides valuable information for diagnosing pediatric TB in remote areas with limited diagnostic testing resources. I have a few minor comments

Line 68 and 69- Please add total number of patients assessed into the methods section

Line 175 Please clarify <=0.00% how can you be less than the “0” percentile?

In Table 2, If there were 66 children assessed by the local TB unit, how come the total number in table 2 aren’t all 66?

Figure 2- what do the different shades and colors mean? A legend defining these would be helpful to readers

Results/Discussion: Maybe an overall recommendation would be a good addition to the manuscript. Based on the comparison, what would be the most ideal criteria for diagnosing pediatric TB in PNG? Especially since the current methods may underdiagnose.

Discussion: how low is BCG uptake? Is this something that is regularly integrated into the diagnostic workup in PNG?

Reviewer 2 Report

Comments and Suggestions for Authors

This manuscript describes the application and comparison of three existing pediatric TB diagnostic approaches/algorithms to improve TB diagnosis in children in the Torres Strait. The use of these tools in remote settings are poorly documented, despite their potential to increase early detection of children with TB. While the aim of this study is worthwhile to pursue, it is not clear that some of the discussion points can truly be gleaned from the small sample size and modified use of each of the approaches.

Introduction:

-Note that “Once diagnosed with presumptive TB..” but fail to state what the definition of that is locally.

-That sentence also makes it seem that a ‘diagnosis of presumptive TB’ is all that is needed locally to commence treatment, is that accurate?

Figure 1:

Figure 1 notes the criteria for TB work up, but it is unclear how the TB work-up is assessed; does the children have to report passively to the health facility and see a clinician / nurse to get this work up? How is it determined which children get this work up? Can there be a line connecting the “minimum criteria” box to the next step to better understand the flow?

Methods:

-Please define presumptive TB.

-Would be beneficial to the reader to more clearly define what signs or symptoms will trigger a diagnostic work-up instead of “Diagnostic work-up … triggered when a child presents with at least one sign or symptoms suggestive of TB.”

-Local flow should be better described, including what is included in the TB work-up and what specimens are collected and how.

-The authors describe how weight loss was collected, but not any other symptoms or characteristics. Can this be described?

-A figure of the local area and facility locations may be beneficial who are unclear about the geographic boundaries; authors discuss crossing the international border and services that can or cannot be conducted in relation.

-It is unclear what the paragraph beginning with “Unfortunately, local policy prevents…” adds related to the clinical decision.

-The manuscript would benefit from more information in the text about the Keith Edwards TB score; at least minimal information for a reader to understand without having to constantly refer to external publications.

-The analyses are unclear; frequencies and proportions of what? Associations between what characteristics?

Results:

Additionally, the sample size is relatively small; reclassifying a few as having / not having TB based on the algorithms will lead to large percent shifts across approaches. This may lead to an inability to ascertain any real valuable results about which approach is best to apply in this setting.

Figure 2: this is very difficult to read and understand. What do the different shades of colors represent?

Figure 3: the authors note that individuals meet or do not meet “TB threshold”, do they mean TB treatment threshold? I see the “note” that the authors left, but instead of this, which is not front and center, it should just be relabeled. Language throughout the document is inconsistent about whether the authors are referring to these algorithms aiding in diagnosis or in identifying individuals who should rapidly initiate treatment. With lack of a gold standard and lab-confirmed TB in all individuals, I think they need to be clear that the accuracy of a TB diagnosis cannot be assessed. It is unclear what this figure adds; it is difficult to read and the same symptoms are carried across all scales.

Discussions:

-Authors refer to the modifications made to the approaches, but these are not fully clear in the methods section.

-Additionally, the sample size is relatively small; reclassifying a few as having / not having TB based on the algorithms will lead to large percent shifts across approaches. This may lead to an inability to ascertain any real valuable results about which approach is best to apply in this setting. This is not mentioned in the limitations section.
